# CrossPT-EEG: A Benchmark for Cross-Participant and Cross-Time Generalization of EEG-based Visual Decoding

## Abstract

Exploring brain activity in relation to visual perception provides insights into the biological representation of the world. While functional magnetic resonance imaging (fMRI) and magnetoencephalography (MEG) have enabled effective image classification and reconstruction, their high cost and bulk limit practical use. Electroencephalography (EEG), by contrast, offers low cost and excellent temporal resolution, but its potential has been limited by the scarcity of large, high-quality datasets and by block-design experiments that introduce temporal confounds. To fill this gap, we present CrossPT-EEG, a benchmark for cross-participant and cross-time generalization of visual decoding from EEG. We collected EEG data from 16 participants while they viewed 4,000 images sampled from ImageNet, with image stimuli annotated at multiple levels of granularity. Our design includes two stages separated in time to allow cross-time generalization and avoid block-design artifacts. We also introduce benchmarks tailored to non-block design classification, as well as pre-training experiments to assess cross-time and cross-participant generalization. These findings highlight the dataset's potential to enhance EEG-based visual brain-computer interfaces, deepen our understanding of visual perception in biological systems, and suggest promising applications for improving machine vision models.

## 1 Introduction

Recent progress has been made in extracting clues from the human brain to inform advancements in AI and neuroscience, largely driven by the extensive use of functional magnetic resonance imaging (fMRI) (Heeger & Ress (2002); Logothetis (2008); Logothetis et al. (2001)) and magnetoencephalogram (MEG) (Benchetrit et al. (2023)) datasets. fMRI and MEG are widely used to investigate various cognitive functions, neurological disorders, and brain connectivity patterns (Antonello et al. (2024); Ye et al. (2024); Toneva et al. (2022); Tang et al. (2024)). Driven by the use of deep neural networks, particularly diffusion-based and transformer-based models, it is even possible to reconstruct human's visual perceptions from fMRI or MEG recordings (Takagi & Nishimoto (2023); Scotti et al. (2024); Ozcelik & VanRullen (2023); Cheng et al. (2023)). These successes are largely attributed to the availability of large-scale datasets, which offer comprehensive data essential to perform extensive studies and in-depth analyses (Richards et al. (2019); Lin et al. (2014)). These models and large-scale datasets have opened new avenues for understanding the brain's intricate functions and for developing advanced applications in brain-computer interfaces, neuroimaging, and beyond (St-Yves et al. (2023)).

In addition to fMRI and MEG, electroencephalography (EEG) is another vital tool in neuroscience research. EEG is easy to use, cost-efficient, and has high temporal resolution, making it a valuable tool for capturing rapid and real-time brain dynamics on the order of milliseconds (Teplan et al. (2002)). EEG signals can be obtained non-invasively by placing electrodes on the scalp, making it a less intrusive method for monitoring brain activity. However, studies on visual perception with EEG signals are limited because of three challenges: (1) the lack of large-scale, high-quality EEG dataset; (2) existing EEG datasets typically featured coarse-grained image categories, lacking fine-grained categories; and (3) many existing datasets use block-design experiments, causing temporal effects that degrade data quality.

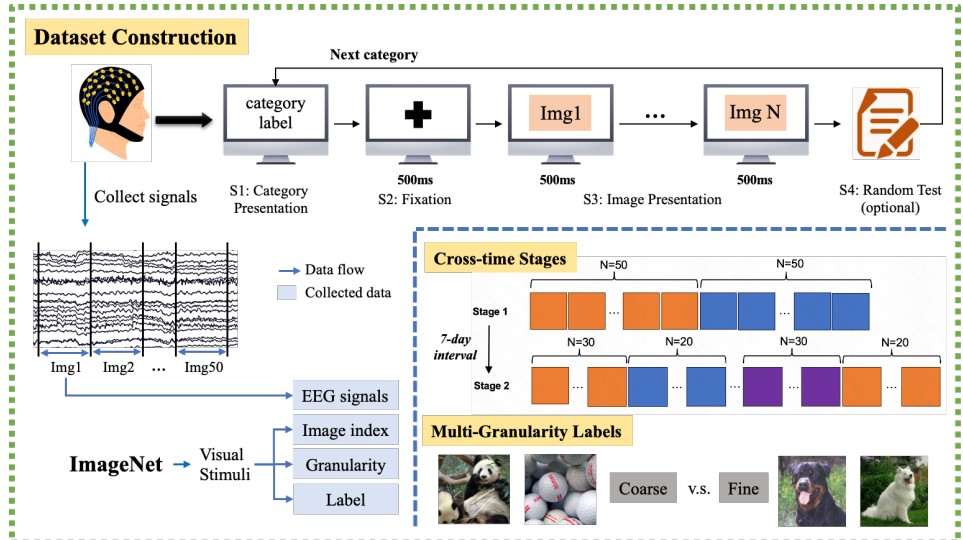

Figure 1: The overall procedure of our dataset construction. The experimental paradigm involves four steps: S1: Category Presentation (displaying the category label), S2: Fixation (500 ms), S3: Image Presentation (each image displayed for 500 ms), and S4: an optional random test to verify participant engagement. The stimuli images are sourced from ImageNet21k, with EEG signals aligned to image indices, granularity levels, and labels. Data flow is indicated by blue arrows, while collected data is highlighted in gray. Stage 2 experiment is conducted after a 7-day interval, adopting a non-block-design experimental paradigm. The same colors represent the same categories.

On the one hand, existing visual-EEG datasets such as Spampinato et al. (2017) are limited by a small number of participants and a restricted set of stimulus images. The limited data volume of current EEG datasets limits research findings' statistical power and generalizability. For example, Things-EEG Gifford et al. (2022) employs a Rapid Serial Visual Presentation (RSVP) paradigm with extremely short image presentation durations (e.g. 50 ms or 100 ms), where only early neural responses are captured, limiting the ability to study the full temporal dynamics of visual perception. On the other hand, the labels in existing EEG datasets are frequently coarse and lack the granularity needed for detailed analysis. Multi-granularity labels are essential because they allow for a more nuanced analysis at different levels of detail. For instance, labels can range from broad categories like "panda" or "golf ball" to more specific attributes like "Rottweiler" or "Samoyed". Third, recent work shows a block-design experimental paradigm, in which EEG signals are affected by temporal effects both before and after stimuli, leading to biases in within-session classification results (Li et al. (2020a)). These challenges underscore the necessity for new visual-EEG datasets and benchmarks that incorporate larger-scale, multi-granularity labels as well as cross-time experimental designs.

To address these challenges, we present *CrossPT-EEG*, a novel EEG benchmark specifically designed to promote research related to visual neuroscience, biomedical engineering, etc. As shown in Figure 1, CrossPT-EEG has a comprehensive dataset that includes EEG recordings from 16 subjects, with a total of 22 sessions, each exposed to 4,000 images sourced from the ImageNet-21k (Ridnik et al. (2021)). These images span 80 different categories, with 50 images per category. The dataset is structured to support multi-granularity analysis, with 40 categories dedicated to coarse-grained tasks and 40 to fine-grained. For 6 of the participants, we conducted Stage 2 experiment after a 7-day interval, adopting a non-block-design experimental paradigm. CrossPT-EEG incorporates classification experiments on non-block-design data, which avoids temporal confounds and thus offers a more reliable evaluation of EEG-based visual decoding. Furthermore, it integrates cross-time experiments that evaluate model generalization across sessions separated by several days, cross-participant experiments that assess robustness across individuals, and pre-training experiments that explore whether leveraging data from other participants can enhance cross-time recognition. Together, these tasks form a comprehensive benchmark designed to evaluate the robustness, generalization, and scalability of EEG-based visual decoding models, offering a solid foundation for both neuroscience and brain–computer interfaces.

Table 1: Detailed metadata for various neurological datasets based on visual stimuli. We only compare stimuli duration when stimuli modalities is image.

| Dataset | #Subjects | Modalities | Visual Stimuli | #Stimuli | Stimuli duration |
|---|---|---|---|---|---|
| GOD (Horikawa & Kamitani (2017)) | 5 | fMRI | ImageNet | 1,200 | 1 s |
| NSD (Allen et al. (2022)) | 8 | fMRI | MS COCO | 10,000 | 3 s |
| DEAP (Koelstra et al. (2011)) | 32 | EEG, ECG | music video (1 min) | 40 | - |
| SEED (Zheng & Lu (2015)) | 15 | EEG | movie clips (4 min) | 15 | - |
| Spampinato et al. (2017) | 6 | EEG | ImageNet | 2000 | 500 ms |
| AMIGOS Miranda-Correa et al. (2018)) | 40 | EEG, ECG | long/short videos | 20 | - |
| Li et al. (2020b) | 6 | EEG | ImageNet, Hollywood 2 | 2000+384 | 500 ms |
| Things EEG2 (Gifford et al. (2022)) | 10 | EEG | Things | 16740 | 100 ms |
| Things EEG1 (Grootswagers et al. (2022)) | 50 | EEG | Things | 22248 | 50 ms |
| EEG-SVRec (Zhang et al. (2024)) | 30 | EEG, ECG | short videos | 2636 | - |
| EIT-1M (Zheng et al. (2024)) | 5 | EEG | CIFAR-10 | 60,000 | 50 ms |
| Alljoined (Xu et al. (2024)) | 8 | EEG | MS COCO | 10,000 | 300 ms |
| **CrossPT-EEG (Ours)** | 16+6 | EEG | ImageNet | 4,000 | 500 ms |

We summarize the main contributions of this work as follows: 1) We propose CrossPT-EEG, the first EEG benchmark with multi-granularity semantic labels, designed for cross-participant and cross-time generalization of visual decoding from electroencephalogram. 2) We significantly scales up existing EEG-visual datasets in the number of subject sessions and total recording duration, providing high-quality and long-duration EEG segments for semantic-level visual understanding. 3) We establish benchmarks including classification tasks on non-block-design EEG data as well as pre-training experiments across time participants, thereby enabling systematic evaluation of model robustness and generalization.

## 2 RELATED WORK

In this section, we review some datasets related to visual recognition and neuroscience and compare them with CrossPT-EEG, as shown in Table 1.

Visual recognition is a cornerstone of computer vision, driven by datasets like ImageNet (Ridnik et al. (2021)), CIFAR (Krizhevsky et al. (2009)), and MS COCO (Lin et al. (2014)). Efforts to combine visual recognition with human have led to datasets like the SALICON dataset (Jiang et al. (2015)), which extends MS COCO with eye-tracking data, enabling the study of visual attention and saliency through large-scale annotations. Neuroscience datasets utilizing fMRI have further enriched this field. The Generic Object Decoding dataset (Horikawa & Kamitani (2017)) captures brain activity while subjects view and imagine objects, facilitating the decoding of mental images. The NSD (Allen et al. (2022)) is a large-scale fMRI dataset in visual neuroscience, recording high-resolution (1.8-mm) whole-brain 7T fMRI data from 8 subjects exposed to 9,000–10,000 color natural scenes from the MS COCO dataset over the course of a year.

fMRI is renowned for its high spatial resolution, allowing researchers to obtain detailed images of brain activity by measuring changes in blood flow (Logothetis et al. (2001)). This capability makes fMRI particularly effective for identifying the specific brain regions involved in various cognitive and sensory tasks. In contrast, EEG offers several distinct advantages over fMRI. EEG is relatively easy to use and cost-efficient, with a straightforward setup that involves placing electrodes on the

scalp to measure electrical activity. One of the most significant benefits of EEG is its exceptional temporal resolution, which captures neural dynamics on the order of milliseconds (Teplan et al. (2002)). This makes EEG ideal for studying fast-occurring brain processes and real-time neural responses, providing insights into the timing and sequence of neural events (Liu et al. (2021)). However, EEG signals collected using portable devices often have a low signal-to-noise ratio, which can complicate data analysis and reduce the accuracy of the results (Kannathal et al. (2005)).

Existing EEG datasets span a variety of research areas. The SEED (Zheng & Lu (2015)) focuses on emotion recognition with detailed EEG recordings from subjects exposed to various emotional stimuli. The BCI Competition IV datasets (Zhang et al. (2012)) provide EEG data for motor imagery tasks, while the TUH EEG Corpus (Shah et al. (2018)) is a large clinical EEG collection often used for benchmarking EEG data quality across different conditions. The DEAP (Koelstra et al. (2011)) collects EEG and peripheral physiological signals from 32 participants as they watch 40 one-minute music videos, providing comprehensive emotional responses. Similarly, the AMIGOS (Miranda-Correa et al. (2018)) captures EEG and physiological responses from participants watching short video clips designed to evoke specific emotional states. In the realm of visual recognition, datasets like the EEG-Classification dataset (Spampinato et al. (2017)) involve 6 subjects viewing 2,000 images across 40 object classes from the ImageNet10k. The Things EEG (Grootswagers et al. (2022); Gifford et al. (2022)) utilizes in the RSVP paradigm features extremely short image presentation durations (50/100 ms) and collects EEG-image pairs from large-scale datasets, making it a valuable resource for research in visual neuroscience. Another significant dataset is the EEG-SVRec (Zhang et al. (2024)), which includes EEG recordings from 30 participants interacting with short videos, aiming to capture detailed affective experiences.

## 3 DATASET CONSTRUCTION

During the data collection process of our user study, participants are presented with a visual stimuli dataset containing 4000 natural images from ImageNet21k. Throughout this process, we continuously record their EEG signals. The whole experimental process is carried out in the laboratory environment. This section describes the entire process of CrossPT-EEG dataset construction.

### 3.1 PARTICIPANTS

We enlist a total of 16 participants via social media, including 10 males and 6 females. These participants are all college students aged between 21 and 27, with an average age of 24.06 and a standard deviation of 1.69. Their majors encompass computer science, mechanical engineering, chemistry, and environmental engineering, and they range from undergraduate to postgraduate levels. All participants are right-handed and assert their proficiency in utilizing image search engines in their daily routines. Each participant dedicates approximately 2 hours to complete the experiment each stage, which includes 30 minutes for equipment setup and task instructions. Before the experiment, participants are informed of a compensation of US$11.8 per hour upon completion, to ensure the quality of the data collected for the study.

### 3.2 STIMULI DATASET

The dataset used for visual stimuli was a subset of ImageNet21k, containing 80 categories of objects. Each category comprises 50 manually curated images, ensuring that each image has a width and height greater than 300 pixels and prominently features an object corresponding to its class label in ImageNet. Additionally, every image is free of watermarks. In this manner, we have selected a total of 4000 high-quality natural images as our visual stimulus dataset.

Among all categories, the first half is consistent with the EEG-Classification dataset (Spampinato et al. (2017)), comprising 40 significantly distinct categories from ImageNet1k. We treat these as *coarse-grained* tasks. The latter 40 categories are designed as a *fine-grained* task, divided into 5 groups with 8 categories each. The categories within the same group share the same parent node in WordNet, and each category label is either a leaf node or a sub-leaf node in WordNet. This selection ensures that the chosen categories represent similar granularity while avoiding overly obscure categories, thereby minimizing potential biases in the experimental results. For instance, coarse-grained categories include items such as African elephants, pandas, mobile phones, golf balls, bananas, and

pizzas. Under the parent node "musical instruments", the fine-grained categories include accordions, cellos, flutes, oboes, snare drums, and trombones.

## 3.3 PROCEDURE

Before engaging in the user study, participants are required to fill out an entry questionnaire and sign a consent about the protection of privacy security. They will receive an orientation regarding the primary tasks and operational procedures. Additionally, they will be notified of their right to withdraw from the study at any point. Before the main trials, participants will undergo a series of training trials designed to familiarize them with the procedures of formal experiments.

Before each experiment, every participant is required to select a unique random seed, which is used to randomize both the order of the categories and the order of images within each category. This randomization is applied in both experimental stages, ensuring a fair distribution of categories and images across participants. For the same participant, a different seed is used for Stage 1 and Stage 2. The experimental-platform code is publicly available in the code repository. The experimental platform follows a sequential and repetitive process as illustrated in Figure 1. (S1) The experimental platform presents the current category label. Participants can proceed to the next step by pressing the space key. (S2) A fixation cross is shown at the center of the screen, ensuring attention is drawn when images are displayed. This fixation period lasts for 500 ms. (S3) The images of this category are sequentially presented using the Rapid Serial Visual Presentation (RSVP) paradigm, which is commonly employed in psychological experiments. Each image is presented for a duration of 500 ms (Kaneshiro et al. (2015)), with a total number of $N$. (S4) Random tests are conducted to verify the participant's engagement in the experiment after the presentation. Specifically, at a random moment after completion of a category block, the participant is asked to answer which category was just presented. Data from categories for which participants fail the test will not be included in final analyses. The EEG signals of the participant will be captured and recorded continuously during the entire process. The program will cycle back to step S1 and display the next category, repeating this process until all the images have been presented.

Our experiment consists of two stages. The first stage follows the same setup as Spampinato et al. (2017) where images from each category are presented consecutively, with $N = 50$ as shown in Figure 1. All of 16 participants took part in this stage. However, as Li et al. (2020a) pointed out, the experimental results under the paradigm of such block-design may be influenced by temporal effects when using shuffled training and test sets. Therefore, we conducted a second stage of the experiment, with $N = 30/20$ and random shuffling. Six participants participated in this phase. Stage 2 was conducted at least seven days after Stage 1. The 7-day interval was intended to prevent potential biases arising from memory effects (Fisher & Radvansky (2018)). Ultimately, the dataset we construct includes the EEG signals of participants exposed to each image visual stimulus in each valid session, along with the corresponding category's wnid and the image's index in ImageNet21k.

## 3.4 DATASET DESCRIPTION

The CrossPT-EEG dataset contains a total of 87,850 EEG-image pairs from 16 participants, with a total of 22 sessions (6 participants took part in two sessions). Each EEG data sample has a size of $(n_{channels}, f_s \cdot T)$, where $n_{channels}$ is the number of EEG electrodes, which is 62 in our dataset; $f_s$ is the sampling frequency of the device, which is 1000 Hz in our dataset; and T is the time window size, which in our dataset is the duration of the image stimulus presentation, i.e., 500 ms. Due to ImageNet's copyright restrictions, our dataset only provides the file index of each image in ImageNet and the wnid of its category corresponding to each EEG segment. Additional information about the dataset is shown in Appendix A.1.

## 4 BENCHMARKS SETTINGS

In this section, we detail the benchmarks of our study by outlining the preprocessing steps, feature extraction methods, task definitions, and models used. The detailed code can be accessed openly through the url `https://anonymous.4open.science/r/EEG-ImageNet-anonymous-74B0`.

Figure 2: Comparison of the training and testing sets used for each task in CrossPT-EEG.

## 4.1 PREPROCESSING

We perform a series of preprocessing steps for the raw EEG data we collect to eliminate noise and artifacts and improve signal quality (Ye et al. (2024)). The preprocessing pipeline includes the following steps: First, re-referencing is done using the offline linked mastoids method, which uses the average of the M1 and M2 mastoid electrodes as the new reference point (Yao et al. (2019)). Then, filtering is performed using a 0.5 Hz to 80 Hz band-pass filter to remove low-frequency drifts and high-frequency noise. Additionally, 50 Hz environmental noise is eliminated. Finally, artifact removal eliminates abnormal amplitude signals and artifacts caused by blinks or head movements.

## 4.2 FEATURE EXTRACTION

In our benchmarks, we extract the 40ms-440ms segment of each EEG signal as the domain feature input. This approach helps to minimize the influence of preceding and subsequent image stimuli on the current stimulus. For models requiring frequency-domain features as input, we extract the differential entropy (DE) of the extracted time-domain signals as features, as this characteristic effectively captures the complexity and variability of brain activity in the frequency domain (Duan et al. (2013)). According to the general division in neuroscience, the frequency bands are categorized as delta (0.5-4 Hz), theta (4-8 Hz), alpha (8-13 Hz), beta (13-30 Hz), and gamma (30-80 Hz). We use the Welch method with a sliding window to estimate the power spectral density $P(f)$ in each frequency band. Then, we normalize the data and calculate the differential entropy (DE) using the formula, $DE = -\int P(f) \log(P(f)) df$. Consequently, for each segment of EEG signals, we obtain the differential entropy (DE) for each electrode and each frequency band.

## 4.3 TASK DEFINITION

To demonstrate the effectiveness of CrossPT-EEG with its cross-time, two-stage design, we conduct EEG-based object classification tasks under different experimental settings. The object classification task is defined as predicting the category of the image stimulus that a participant is viewing, based solely on their EEG signals.

It is important to note that block-design (all images of the same category are presented together) experiments can introduce temporal markers into EEG data, which may artificially inflate classification performance when evaluated only within Stage 1. While such results are not representative of

real-world generalization, they may still provide useful insights for future research on EEG temporal effects. For completeness, we include these Stage 1-only results in the Appendix A.4.

We conduct evaluations on EEG-based visual decoding with four tasks, **Within-Time** (WT), **Cross-Time** (CT), **Cross-Participant** (CP), and **Pre-training** (PT). As shown in Figure 2, WT using 30 images per category for training and 20 images per category for testing, both from Stage 2. As WT adopted a non-block-design paradigm in which images used for training and testing are temporally separated, the results are free from class-specific temporal bias. In the CT task, the training data come from Stage 1 of a given participant, while the testing data come from Stage 2 of the same participant. Thus, CT evaluates cross-time generalization within the same individual. In the CP task, the training data also come from Stage 1, but specifically from the 10 participants who did not participate in Stage 2. The testing data comes from Stage 2 of the remaining participant. Thus, CP evaluates generalization across participants as well as across sessions. Finally, we investigated cross-subject, cross-time pre-training experiments (PT) to investigate whether pre-training on other participants' data could improve recognition performance in cross-time classification. This setting enables us to examine the benefits of leveraging additional subjects for pre-training in enhancing the robustness and generalization of visual decoding models. On our multi-granularity labeled image dataset, we test the above tasks with different levels of granularity.

In our experiments, we employed two-way identification as the primary evaluation metric (Ozcelik & VanRullen, 2023). We chose this measure because it provides a clearer basis for cross-task comparisons. For completeness, we also report additional metrics, such as accuracy, in the Appendix A.4. In two-way identification, for each test sample, we randomly select k comparison samples from different categories. The model is then required to decide which of the two samples is more likely to belong to the ground-truth class. Each decision is scored as 1 if correct and 0 otherwise. Final performance is reported as the average over all pairwise comparisons, with a chance level of 0.5. In our benchmarks, we set k = 500 to ensure stable and robust evaluation.

### 4.4 MODELS

We employ simple machine-learning classification models such as ridge regression, KNN, random forest, and SVM. Additionally, we implement deep learning models including MLP, EEGNet (Lawhern et al. (2018)) and RGNN (Zhong et al. (2020)). These models are the most commonly used and have demonstrated excellent performance in EEG-related research. We train the models on a single NVIDIA 4090 GPU. For model and training details, please refer to the code and Appendix A.3.

## 5 EXPERIMENTAL RESULTS

Table 2: The average results of all participants in the coarse-grained classification task. * indicates the use of time-domain features, otherwise the use of frequency-domain features. † indicates that the difference compared to random is significant with p-value $< 0.05$.

| Model | | WT | CT | CP | PT |
|---|---|---|---|---|---|
| Classic model | Ridge | 0.861±0.022† | 0.509±0.011 | 0.501±0.007 | 0.508±0.010 |
| | KNN | 0.892±0.026† | 0.517±0.019 | 0.491±0.006 | 0.521±0.017 |
| | RandomForest | 0.909±0.031† | 0.511±0.019† | 0.507±0.011 | 0.524±0.020† |
| | SVM | 0.935±0.039† | 0.536±0.018† | 0.506±0.015 | 0.544±0.017† |
| Deep model | MLP | **0.938±0.024**† | 0.552±0.012† | 0.524±0.016† | 0.570±0.020† |
| | EEGNet* | 0.878±0.022† | 0.519±0.015† | 0.509±0.012 | 0.524±0.022† |
| | RGNN | 0.933±0.025† | **0.566±0.015**† | **0.526±0.018**† | **0.585±0.027**† |

The coarse-grained represents the 40-class classification accuracy; and the fine-grained represents the average accuracy of five 8-class classification tasks. Table 2 and Table 3 shows the average results of all participants in the coarse-grained and grained classification task, respectively. We observe that in the within-task (WT) task, both classic models and deep models achieve strong performance, indicating that WT is the relatively easiest among the four tasks. The best-performing model reaches a two-way identification close to 0.95, suggesting that EEG signals contain sufficient information

Table 3: The average results of all participants in the fine-grained classification task. * indicates the use of time-domain features, otherwise the use of frequency-domain features. † indicates that the difference compared to random is significant with p-value < 0.05.

| | Model | WT | CT | CP | PT |
|---|---|---|---|---|---|
| Classic model | Ridge | 0.886±0.023† | 0.549±0.017† | 0.495±0.007 | 0.543±0.012† |
| | KNN | 0.909±0.025† | 0.554±0.017† | 0.505±0.010 | 0.551±0.015† |
| | RandomForest | 0.922±0.030† | 0.587±0.022† | 0.511±0.012 | 0.595±0.019† |
| | SVM | 0.949±0.035† | 0.592±0.021† | 0.509±0.019 | 0.606±0.022† |
| Deep model | MLP | **0.954±0.026**† | **0.617±0.019**† | 0.563±0.026† | **0.636±0.027**† |
| | EEGNet* | 0.893±0.023† | 0.566±0.027† | 0.522±0.026† | 0.579±0.025† |
| | RGNN | 0.945±0.024† | 0.601±0.024† | **0.573±0.028**† | 0.634±0.027† |

for reliable decoding when training and testing are conducted within the same stage. However, in the cross-time (CT) task, performance drops substantially across all models. This degradation is particularly obvious for classic models, which primarily rely on learning fixed linear mappings. This result highlights the challenge posed by temporal variability in EEG signals. In the cross-participant (CP) task, the performance of classic models is further reduced, in some cases barely above chance, suggesting that these methods fail to generalize across individuals. By contrast, deep models consistently deliver better results across tasks. This indicates that graph-based neural networks are able to capture more stable and transferable patterns from EEG signals, making them promising candidates for addressing inter-individual differences. Meanwhile, the relatively poor performance of EEGNet may be attributed to the fact that the original version of EEGNet was not designed for semantically related tasks. Finally, in the pre-training (PT) task, deep models again show clear advantages, with RGNN and MLP achieving the highest performance across coarse-grained and fine-grained tasks. In the PT task, deep models outperform their CT counterparts across different levels of granularity, suggesting that leveraging pre-training with data from other participants can effectively improve cross-time recognition.

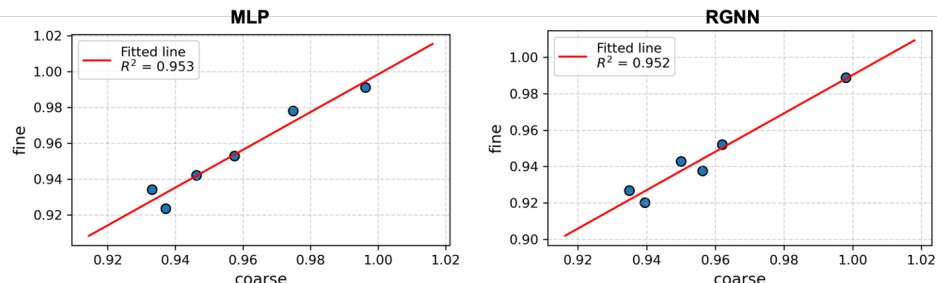

Figure 3: MLP and RGNN performance of the "coarse" (vertical axis) and "fine" task (horizontal axis) in WT task.

To better compare the differences between "fine" and "coarse" tasks, we randomly select 8 coarse-grained categories and perform WT task because the result is the best on this task. This process is repeated 5 times, and the results for each participant is calculated and plotted alongside their average "fine" task results in Figure 3. We then perform linear regression on the data points, and the resulting function has a slope greater than 1, indicating that models generally achieve better results on the "coarse" classification tasks. This finding is also consistent with intuition.

We further aggregate results of different participants across different tasks for comparison, as shown in Figure 4. MLP and RGNN exhibit remarkably similar performance distributions across different tasks under both coarse and fine granularity settings. Both models follow the same trend ordering: WT ≫ PT > CT > CP. This indicates that within-time classification remains the easiest task, while cross-participant is consistently the most challenging. This trend aligns with prior works that have compared cross-subject and cross-session EEG decoding, which often report that the most difficult scenario is generalization across participants (Liang et al., 2022; Chen et al., 2021; Apicella

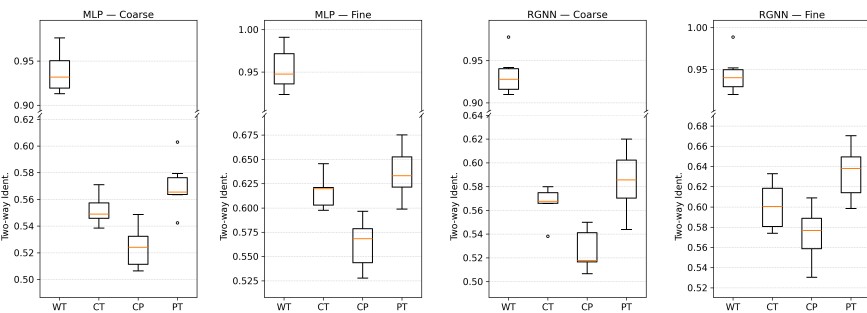

Figure 4: Cross-Task comparison of MLP and RGNN performance with different granularity levels.

et al., 2024). Our results not only corroborate this conventional wisdom but also further demonstrate that pre-training on data from other participants can significantly boost object classification performance relative to direct cross-time generalization.

## 6 DISCUSSION AND CONCLUSION

In this paper, we introduced CrossPT-EEG, a novel EEG benchmark designed to advance research in visual neuroscience. CrossPT-EEG has a comprehensive and richly annotated dataset, comprising EEG recordings from 16 subjects, with a total of 22 sessions exposed to 4000 images from 80 different categories. We further establish benchmarks that encompass classification tasks on non-block-design EEG data and pre-training experiments spanning both time and participants. These benchmarks provide a systematic framework for evaluating the robustness and generalization of visual decoding models, and highlight the potential to advance cross-time, cross-subject learning in both cognitive neuroscience and brain–computer interface applications.

**Limitation**. Firstly, while our dataset is more comprehensive than similar works, each participant's data is still relatively limited. This necessitates the development of inter-subject models to overcome this limitation and enhance generalizability. Secondly, it is limited in representation, as participants were drawn from a specific sample. This results in an age distribution skewed towards teenager individuals and a racial composition predominantly White and Asian. Future work should aim to include a more diverse and extensive participant pool. Additionally, although our non-block-design paradigm and cross-time training strategy circumvent the temporal effects that often confound block-design experiments, we did not conduct an in-depth investigation of the temporal effects present in Stage 1. Future work could leverage our dataset to explore this direction more systematically. Lastly, our benchmarks did not incorporate many of the latest deep-learning methods. Instead, we employed a set of commonly used models primarily to demonstrate the effectiveness and to illustrate how our two-stage, cross-time dataset can be utilized. We believe that recent advancements in deep learning could greatly benefit from our comprehensive dataset, potentially leading to significant breakthroughs in visual neuroscience.

**Insight for ML**. CrossPT-EEG provides a comprehensive resource for developing models in visual recognition tasks, enabling the development of sophisticated deep-learning models capable of capturing intricate patterns within EEG data. Future research could leverage the dataset to enhance domain adaptation and transfer learning techniques, facilitating effective inter-subject and cross-time task completion. By offering a diverse dataset of visual stimuli and supporting multi-level classification tasks, CrossPT-EEG could foster the creation of hierarchical models that mirror human cognitive processes and improve the generalization capabilities of machine learning algorithms. We hope this benchmark will enable the development of more sophisticated models and methodologies, driving forward EEG-based visual neuroscience research and offering deeper insights into the neural mechanisms underlying visual perception and processing.

**Insight for BCI**. As hardware technology progresses, portable EEG devices are becoming increasingly feasible, offering new opportunities for real-time BCI applications. Researchers could develop

robust BCI systems that accurately interpret user intent from EEG signals. The comprehensive size and diverse visual stimuli in CrossPT-EEG allow for the creation of adaptive BCI systems that learn and respond to long-term individual user patterns. This paves the way for personalized neurotechnology solutions, particularly enhancing human-computer interaction for individuals with disabilities. Furthermore, addressing privacy protection and ethical concerns will be crucial as BCI technology advances, ensuring user data is securely handled and individual rights are respected.

## 7 ETHICS AND PRIVACY

To protect participants' privacy and physical health, our user study adheres to strict ethical guidelines for human research, with approval from the ethics committee[1]. In accordance with ethical standards, we have taken several steps to protect participants' privacy, including data anonymization and obtaining informed consent from all participants. Participants were fully briefed on the study's aims, procedures, and potential implications. Moreover, the EEG recording procedure used is entirely non-invasive and involves no risk to participants.

## 8 REPRODUCIBILITY STATEMENT

We are committed to full reproducibility of this work. The dataset in CrossPT-EEG will be publicly released after the review stage. All code, including data preprocessing, model training, and evaluation scripts, will be made available at github link. The architectural and implementation details of all models are documented in the Appendix A.3.

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

# A APPENDIX

## A.1 ADDITIONAL INFORMATION ABOUT DATASET

The specific statistics of the dataset are shown in Table 4.

Table 4: The Statistics of CrossPT-EEG Dataset.

|  | #Categories | #Images | #Subjects | #EEG-image pairs | Datasize |
|---|---|---|---|---|---|
| CrossPT-EEG | 80 | 4000 | 16 | 87,850 | 20.85GB |

As shown in Listing 1, the CrossPT-EEG dataset storage format is provided after review. The dataset can be accessed through the cloud storage link available in our GitHub repository after review. Due to file size limitations on the cloud storage platform, we split the dataset of Stage 1 into two parts: "CrossPT-EEG_1.pth" and "CrossPT-EEG_2.pth". Users can choose to use only one of the parts based on their specific needs or device limitations. Demographic information is also provided at the file level.

```
{
    "dataset": [
        {
            "eeg_data": torch.tensor,
            "granularity": "coarse"/"fine",
            "subject": 15,
            "label": 'ne2106550',
            "image": 'n02106550_1410.JPEG',
            "stage": 30/20, (This attribute only appears in Stage 2)
        }, ...
    ],
    "labels": [
        "n02106662", ...
    ],
    "images": [
        "n02106662_13.JPEG", ...
    ]
}
```

Listing 1: CrossPT-EEG dataset format.

## A.2 APPARATUS

All the image stimuli are presented on a desktop computer that has a 27-inch monitor with a resolution of 2,560×1,440 pixels and a refresh rate of 60 Hz. Participants are required to use the keyboard to interact with the platform. EEG signals are captured and amplified using a Scan NuAmps Express system (Compumedics Ltd., VIC, Australia) and a 64-channel Quik-Cap (Compumedical NeuroScan). A laptop computer functions as a server to record EEG signals and triggers using Curry8 software. Throughout the experiment, electrode-scalp impedance is maintained under 50$\Omega$, and the sampling rate is set at 1,000Hz.

## A.3 EXPERIMENTAL SETUP DETAILS

We conduct experiments under three different granularity settings: the "all" task includes all 80 categories; the "coarse" task includes 40 coarse-grained categories; and the "fine" task includes 8 fine-grained categories that belong to the same parent node, with the average accuracy calculated across 5 groups.

The model structures and hyperparameters are as follows. For SVM, we try linear, polynomial, and radial basis function (RBF) kernels. The regularization parameter is tested from values $\{10^{-3}, 10^{-2}, 10^{-1}, 1, 10^1, 10^2, 10^3\}$. For RandomForest, we try to set the number of trees in the

forest from values $\{20, 50, 100, 200, 500\}$, with all other parameters set to their default values. For KNN, we set the number of neighbors to $\{5, 10, 15, 20\}$. For ridge regression, all parameters are set to their default values. For RGNN, when calculating the edge weights between electrodes, we use the hardware parameters of our data collection device to determine the topological coordinates of each electrode. In addition to the standard implementation, we add two batch normalization layers. The main hyperparameters adjusted are the number of output channels of the graph convolutional network (i.e., the hidden layer dimension) and the number of hops (i.e., the number of layers). These are set to $\{100, 200, 400\}$ and $\{1, 2, 4\}$ respectively. For EEGNet, we use the standard implementation and set the length of the first step convolution kernel to half the number of sampling time points, which is 200. The main hyperparameters adjusted are the number of output channels for the first convolutional layer (F1) and the depth multiplier (D), which are set to $\{8, 16, 32\}$ and $\{2, 4, 8\}$ respectively. For MLP, we set two hidden layers with dimensions of 256 and 128, respectively. Each linear layer is followed by a batch normalization layer and a dropout layer with a probability of 0.5. In the PT task, during the pre-training phase we train with half of the learning rate for half of the epochs.

For all deep models, we use the cross-entropy loss function. In MLP and EEGNet, we use the SGD optimizer with learning rate $10^{-3}$, weight decay $10^{-3}$, and momentum 0.9, training for 2000 epochs. After that, we adjust the learning rate to $10^{-4}$ and weight decay to $10^{-4}$ and continue training for another 1000 epochs. In RGNN, we use the Adam optimizer with learning rate $10^{-3}$ and weight decay $10^{-3}$, training for 2000 epochs. Subsequently, we adjust the learning rate to $10^{-4}$ and weight decay to $10^{-4}$ and train for an additional 1000 epochs. The batch size is uniformly set to 80.

All the implementations mentioned above are open-sourced and available in the GitHub repository.

### A.4 ADDITIONAL EXPERIMENTAL RESULTS

Table 5 shows the average results of all participants in Stage 1 and Table 6 shows the average results of all participants in Stage 2.

Table 5: The average results of all participants in the object classification task of Stage 1. * indicates the use of time-domain features, otherwise the use of frequency-domain features. † indicates that the difference compared to the best-performing model is significant with p-value $< 0.05$.

| | Model | Acc (all) | Acc (coarse) | Acc (fine) | F1 (all) | F1 (coarse) | F1 (fine) |
|---|---|---|---|---|---|---|---|
| Classic model | Ridge | 0.286±0.074† | 0.394±0.081† | 0.583±0.074† | 0.261±0.070† | 0.373±0.082† | 0.610±0.121† |
| | KNN | 0.304±0.086† | 0.401±0.097† | 0.696±0.068† | 0.286±0.081† | 0.380±0.096† | 0.717±0.132† |
| | RandomForest | 0.349±0.087† | 0.454±0.105† | 0.729±0.072† | 0.323±0.083† | 0.425±0.099† | 0.723±0.092† |
| | SVM | **0.392±0.086†** | **0.506±0.099†** | **0.778±0.054†** | **0.378±0.083†** | **0.486±0.105†** | **0.770±0.054†** |
| Deep model | MLP | 0.404±0.103† | **0.534±0.115** | **0.816±0.054** | 0.397±0.100† | **0.523±0.108** | **0.819±0.053** |
| | EEGNet* | 0.260±0.098† | 0.303±0.108† | 0.365±0.095† | 0.251±0.095† | 0.291±0.098† | 0.374±0.102† |
| | RGNN | **0.405±0.095** | 0.470±0.092† | 0.706±0.073† | **0.401±0.098** | 0.455±0.087† | 0.723±0.079† |

Table 6: The average results of all participants in the object classification task of Stage 2. * indicates the use of time-domain features, the use of frequency-domain features. † indicates that the difference compared to the best-performing model is significant with p-value $< 0.05$.

| | Model | Acc (all) | Acc (coarse) | Acc (fine) | F1 (all) | F1 (coarse) | F1 (fine) |
|---|---|---|---|---|---|---|---|
| Classic model | Ridge | 0.182±0.053† | 0.253±0.074† | 0.431±0.108† | 0.178±0.052† | 0.243±0.075† | 0.438±0.107† |
| | KNN | 0.220±0.081† | 0.310±0.113† | 0.574±0.119† | 0.211±0.083† | 0.299±0.105† | 0.565±0.134† |
| | RandomForest | 0.268±0.101† | 0.358±0.129† | 0.609±0.136† | 0.259±0.098† | 0.341±0.117† | 0.596±0.139† |
| | SVM | 0.281±0.090† | 0.368±0.107† | 0.657±0.140† | 0.271±0.084† | 0.365±0.109† | 0.648±0.134† |
| Deep model | MLP | 0.297±0.093 | 0.395±0.110 | **0.718±0.149** | 0.285±0.087 | **0.392±0.108** | **0.710±0.140** |
| | EEGNet* | 0.169±0.044† | 0.244±0.095† | 0.377±0.096† | 0.160±0.041† | 0.228±0.088† | 0.372±0.096† |
| | RGNN | **0.302±0.097** | **0.401±0.105** | 0.693±0.140† | **0.297±0.100** | 0.388±0.106 | 0.701±0.142† |

Table 7 shows the performance of the best-performing participant across all models and tasks of Stage 1..

Figure 5 shows the accuracy for each participant in the object classification task of Stage 1 across SVM, MLP, and RGNN models. We find that the ranking of participants' accuracy is relatively consistent across different models.

Table 7: The best results of all participants in the object classification task of Stage 1.

| Model | | Acc (all) | Acc (coarse) | Acc (fine) |
|---|---|---|---|---|
| Classic model | Ridge | 0.4550 | 0.5375 | 0.7200 |
| | KNN | 0.5025 | 0.6063 | 0.8013 |
| | RandomForest | 0.5006 | 0.6488 | 0.8450 |
| | SVM | **0.5794** | **0.7038** | **0.8588** |
| Deep model | RGNN | **0.6088** | 0.6525 | 0.8050 |
| | EEGNet* | 0.4413 | 0.5213 | 0.5988 |
| | MLP | 0.5925 | **0.7413** | **0.8875** |

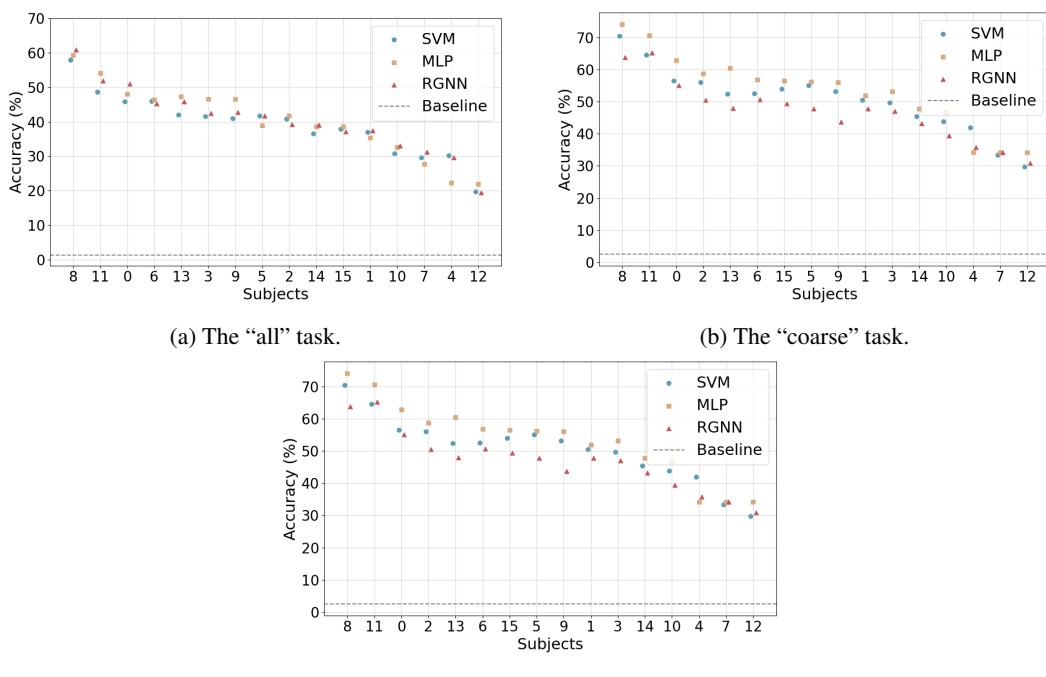

(a) The "all" task.

(b) The "coarse" task.

(c) The "fine" task.

Figure 5: Acc for each participant in the object classification task of Stage 1 across SVM, MLP, and RGNN Models.

## A.5 TEMPORAL EFFECT

In Figure 6, we plotted the average classification accuracy for images at different index positions in the test set under various training and test set splits to show the temporal effect in Stage 1.

We observed that the first few images in the test set have significantly higher accuracy, indicating a strong temporal effect.

## A.6 THE USE OF LARGE LANGUAGE MODELS

In this work, we leveraged large language models (LLMs) to assist in manuscript preparation, including refining the text for clarity and style, as well as facilitating literature retrieval. All LLM-generated suggestions were carefully reviewed, edited, and integrated by the authors to ensure scientific accuracy and consistency with our own writing voice. We acknowledge the ongoing discourse around the ethical use of LLMs in scholarly writing—particularly regarding transparency, originality, and accountability. We transparently report the use of LLM assistance and reaffirm that all substantive intellectual contributions (e.g. experimental design, data analysis, interpretation) originated from the authors.

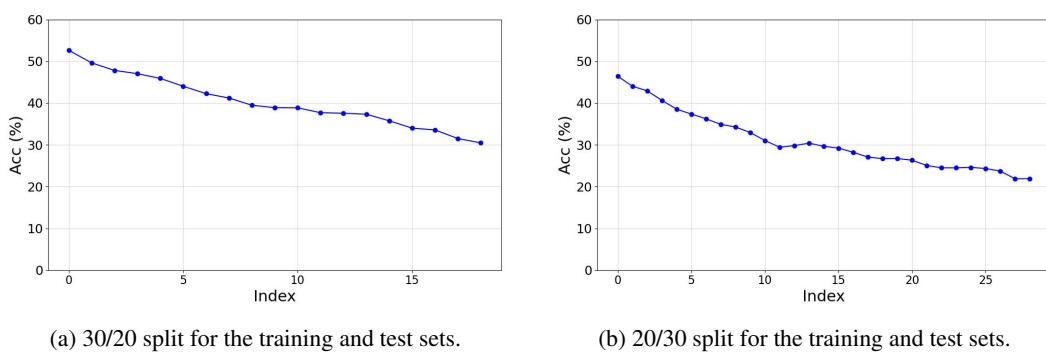

(a) 30/20 split for the training and test sets.    (b) 20/30 split for the training and test sets.

Figure 6: Average classification accuracy under different training and test set splits, with accuracy plotted against the indices of image stimuli in the test set.

