# OpenReview forum: "EEG-ImageNet: A Benchmark for Pre-training and Cross-Time Generalization of EEG-based Visual Decoding"
_ICLR.cc/2026/Conference — Submitted to ICLR 2026_

### Official Review · Reviewer_tGfm · 2025-10-25

**Soundness:** 1
**Presentation:** 1
**Contribution:** 1
**Rating:** 0
**Confidence:** 5

**Summary:**

This paper introduces a new dataset for the problem of decoding image stimulus class from EEG recordings. It is similar in nature to the Perceive dataset introduced in Spampinato et al. (CVPR 2017). There are 80 classes (instead of 40 in Perceive). The first 40 are coarse grain, just as in Perceive. The second 40 are fine grain, with 5 superordinate classes of 8 subordinate classes each. Just as in Perceive, there are 50 stimuli per class. While Perceive had 6 subjects, here there are 16. Just like Perceive, stimuli were presented for 500ms. Unlike Perceive, which recorded with an 128 electrode recorder, EEG was recorded from an unspecified 62-electrode recorder. Just like in Perceive, stimuli were presented in blocks, where all stimuli in the block were of the same class and all stimuli of that class were in the same block. Unlike Perceive, each block started with a presentation of the class label, presumably as visual text, though unspecified in the paper. Unlike Percevie, each block ended with some sort of test to measure attention, though the nature of this is unspecified in the manuscript, whcih says it was optional. Unlike Perceive, there were two recording session for 6 of the 16 subjects. In the second session, each block of 50 stimuli started with 30 stimuli from one class and ended with 20 stimuli of a different class. Results of classifying this dataset with various models are presented. The central claim is that the two-session design avoids the block confound discussed in Li et al. (2021).

**Strengths:**

None.
The dataset suffers from a known published confound that correlates stimulus class with drift in the EEG signal, essentially an embedded clock. Thus, decoders can and do classify the clock, not stimulus class, as demonstrated by Li et al. (2021) and follow on papers in TPAMI and CVPR by the same authors. As well as Xu et al. (2026) The impacts of temporal autocorrelations on EEG decoding, Biomedical Signal Processing and Control, 113.

**Weaknesses:**

1. Li et al. (2021) subject 6 shows that the block confound even occurs when the training and test sets come from different blocks from different sessions. Thus the 2-session design does not remove the confound. Thus, this dataset still suffers from the block confound and the results are thus not to be trusted. Li et al. (2021) show that performance drops from near perfect to near chance when the confound is removed with randomized trials. There is no excuse that can justify using a block design instead of randomized trials.
 2. Numerous details discussed above are missing, like what the exclusion criterion based on the attention test was. Excluding some trials from some subjects breaks the counterbalanced design. This introduces bias into the classification task and thus chance is not 1/k for k classes. It is not clear this was taken into account in computing statistical significance since the process for computing p values was not discusses. It is not clear whether correction for multiple comparisons was performed.
 3. The stimulus presentation order was not specified. Exactly what was and was not randomized was not specified. Without this, it is impossible to assess the claim that the design does not exhibit correlation between stimulus class and a clock embedded into the signal.
 4. The four tasks, WT, CT, CP, and PT are not described in sufficient detail to understand precisely what was done.
 5. While you claim that this dataset is large, the published datasets associated with Li et al. (2021) and Ahmed et al. (2021)

https://ieee-dataport.org/open-access/dataset-perils-and-pitfalls-block-design-eeg-classification-experiments
https://ieee-dataport.org/open-access/dataset-object-classification-randomized-eeg-trials

are just as large, if not larger, do not suffer from the block confound, and cover video stimuli as well as image stimuli, but are not mentioned or cited.
 6. Many people refer to the Perceive dataset from Spampinato et al. (2017) ad EEG-ImageNet. Other datasets have been collected and published under the name EEG-ImageNet. Reusing the name is confusing.

**Questions:**

1. Why did you not simply conduct randomized trials? That is the standard method universally adopted in all of experimental science to avoid confounds.
2.  Why did you not try your methods on the two datasets mentioned above as these were collected with randomized trials and thus do not suffer from the block confound?

---

> ### Author Response · Authors · 2025-11-25
>
> 1. **Regarding the temporal-effect concern.**
>
> We would like to refer the reviewer to our official unified comment submitted on Nov 13, where we addressed this issue in detail.
> The essential clarification is that our current experimental paradigm does not suffer from the temporal-effect issue identified in previous block-design studies.
> The setting of Li et al. (2021) Subject 6, you mentioned, is in fact most similar to our CT task, not our WT setting, because it evaluates cross-session generalization.
> Our CT results are **fully consistent** with the observations reported in Li et al. (2021). When temporal adjacency is fully removed, performance decreases substantially.
> Therefore, rather than contradicting our conclusions, the Li et al. findings align with and further validate our task-specific results.
>
> 2. **Regarding the attention-test exclusion criterion.**
>
> After some random category blocks, subjects were prompted with a **forced-choice question asking which category had just been presented**. If answered incorrectly, that category’s trials were excluded.
> Only two subjects in Stage 1 had excluded categories (one subject excluded 1 category, another excluded 3 categories).
> This results in the final total of 87,850 EEG–image pairs reported in Appendix A.1.
> Importantly, this exclusion does not affect the chance level of our evaluation, because our primary metric is **two-way identification**, for which the chance level is strictly 0.5, independent of the number of categories. Thus, the exclusion does not introduce bias into the statistical interpretation of the results.
> In addition, all reported p-values are corrected using the **FDR correction**.
> We have revised our paper by adding a full explanation for the attention test in Section 3.3 of the revised manuscript.
>
> 3. **Regarding trial randomization.**
>
> In both stages of the experiment, we randomized **the order of categories and the order of images within each category**.
> Each subject was assigned a unique random seed for stimulus ordering.
> This ensures full randomization of sequence-level factors while maintaining the semantic structure required for eliciting meaningful EEG responses.
> We have now expanded Section 3.3 to clearly specify what was randomized and at which levels, and we also note that **the code for the randomization is also available in the provided repository link**.
>
> 4. **Regarding Figure 2 and task definitions.**
>
> We acknowledge that the previous presentation of the four tasks (WT, ST, CT, PT) may not have been sufficiently clear, which may have contributed to the reviewer’s concerns.
> We have therefore **updated Figure 2 with additional annotations and detailed explanations**, making explicit how each task defines its train/test split and stimulus ordering.
> These revisions ensure that readers can clearly see how the design avoids temporal bias in all tasks.
>
> 5. **Regarding comparisons with other datasets.**
>
> Thank you for pointing this out.
> We have **added comparisons with the datasets mentioned** by the reviewer to the Related Work section.
>
> 6. **Regarding the dataset name.**
>
> We appreciate the reviewer’s suggestion and, together with Reviewer DHLu’s comment, we agree that reusing the term “EEG-ImageNet” may cause unnecessary confusion given existing datasets with similar names.
> We have therefore decided to rename our benchmark to **CrossPT-EEG** and have applied this change **consistently throughout the revised manuscript**.
>
> Thank you very much for your reviews and comments on our paper. If you have any further questions or concerns, we would be happy to continue the discussion. If you find that this response satisfactorily addresses your concerns, we kindly ask you to consider adjusting your review scores accordingly.

---

### Official Review · Reviewer_DHLu · 2025-10-26

**Soundness:** 2
**Presentation:** 2
**Contribution:** 3
**Rating:** 4
**Confidence:** 5

**Summary:**

This paper introduces an EEG dataset collected from 16 participants using ImageNet-based visual stimuli across two experimental stages. Although the dataset offers potential contributions to brain decoding research, several methodological and reporting concerns must be resolved before the work is suitable for publication.

**Strengths:**

See questions

**Weaknesses:**

See questions

**Questions:**

1. The authors acknowledge the temporal confounds introduced by block-design paradigms, yet paradoxically still adopt such a design in both Stage 1 and Stage 2. Although Stage 2 is claimed to be “non-block,” the structure of the experiment still presents images from the same category in short temporal clusters, failing to achieve full randomization. This undermines the core motivation of the dataset, which is to address block-related artifacts. Moreover, if the intention was to retain Stage 1 in order to investigate temporal effects introduced by block design, the paper does not provide any systematic analysis of such effects. Instead, the two stages adopt markedly different protocols (e.g., temporal spacing), thereby introducing an uncontrolled source of variability that further complicates interpretation and comparison across tasks.
2. The procedural description of the experiment is vague and incomplete. Critical details such as how many blocks each participant completed, how long each block lasted, and whether the data was collected in a single continuous session or across multiple sessions remain unspecified. The paper also lacks a clear account of the train-test split strategy: it is unclear whether there is a fixed test set or if different splits are used for each evaluation. This ambiguity is compounded in Figure 2, where it is not indicated whether the training and testing sets in WT, CT, CP, and PT originate from Stage 1, Stage 2, or both. As a result, the methodological transparency is insufficient for reproducibility or proper evaluation of the reported benchmarks.
3. The comparability of the reported results across different tasks is undermined by imbalanced subject participation. Specifically, ten of the sixteen participants did not take part in Stage 2, which directly affects the validity of comparisons across WT, CT, CP, and PT tasks. Because these tasks are evaluated on different subsets of participants, the results are not strictly comparable, yet the paper presents them side by side in Tables 2 and 3 without accounting for this discrepancy. This undermines claims about the relative difficulty or generalizability of each task.
4. The dataset's scale is a notable improvement over previous EEG-visual studies, but it still falls short of the scale required for training large neural models or for making strong claims about cross-subject generalization. With only six participants contributing Stage 2 data and a total of 16 in Stage 1, the participant pool is very limited both in size and age diversity. The authors should moderate their claims of generalization, as the dataset’s limited scale compared to what the name EEG-ImageNet implies suggests that data from a broader and more diverse population is needed to support such claims.
5. The term “cross-time” is used throughout the manuscript, but Stage 2 is conducted at least seven days after Stage 1, making “cross-day” a more precise and appropriate description.
6. The paper evaluates several classic and relatively simple deep models but does not benchmark against recent and widely adopted architectures in EEG decoding. Incorporating mainstream models would substantially strengthen the benchmark’s value, improve reproducibility, and enable more meaningful comparisons with the current state of the art.
7. The manuscript does not include any visualizations of the learned EEG features or model activations, missing an opportunity to underscore the neuroscientific relevance of the signals being classified. Without visual inspection of the EEG responses across categories or tasks, it remains unclear whether the models are learning meaningful neural correlates of perception or simply exploiting low-level artifacts such as EMG or eye movement signals. This data validation is particularly important for a dataset intended to bridge neuroscience and machine learning.
8. While the dataset is described as having coarse- and fine-grained categories, the manuscript provides no visual or conceptual overview of the category structure. Including representative images, a hierarchical map of categories, or example trials would help the reader better understand the nature of the visual stimuli and the complexity of the decoding task.
9. The feature extraction procedure involves a fixed window after stimulus onset, but the analysis does not discuss or control for individual variability in visual latency (e.g., typical VEP components like P1 or N170). Without evaluating how shifts in latency affect decoding performance, there remains a risk of overlap between responses to adjacent images, especially given the 500 ms stimulus duration in the visual presentation paradigm. This could lead to response contamination across trials, confounding model performance.
10. Figure 6 presents image generation results, yet the method used for image reconstruction is not described in sufficient detail. It remains unclear how EEG features were mapped into image space, what generative model was used, how the outputs were evaluated, and what the overall goal of this analysis was.

---

> ### Author Response · Authors · 2025-11-29
>
> Q1. As we stated in our official comment submitted on Nov 13, **the concern on temporal effects no longer applies under our current experimental paradigm**. In that reply, we also explained why we did not adopt a fully randomized stimulus order. We retain the Stage 1 data solely as a training set (see the revised Figure 2). As shown in Section 3.3 and Figure 1, both Stage 1 and Stage 2 follow exactly **the same protocol (including the same time intervals)**, hence there is no source of uncontrolled variability between stages. We hope to avoid your misunderstanding through our clarification.
>
> Q2. In the revised version, we have provided a more detailed description of the experimental protocol. Concretely: **the number of blocks completed by each participant corresponds to the total number of stimulus categories listed in Section 3.2; the duration of each block is explicitly indicated in the upper half of Figure 1. The data were collected in a single continuous experimental session.** We have also updated Figure 2: for each task, we added more annotations and explanations, so that readers can clearly understand how the train/test split and stimulus ordering work together to avoid temporal bias.
>
> Q3. The misunderstanding seems to stem from Figure 2 in the previous version. In the revised version of Figure 2, it is clearly shown that all test sets are drawn from Stage 2 data, so there is **no imbalance in participant numbers across tasks** due to different participation in Stage 1 vs Stage 2.
>
> Q4. We appreciate the reviewer’s suggestion and, together with Reviewer tGfm’s comment, we agree that reusing the term “EEG-ImageNet” may cause unnecessary confusion given existing datasets with similar names.
> We have therefore decided to rename our benchmark to **CrossPT-EEG** and have applied this change consistently throughout the revised manuscript.
>
> Q5. In the EEG research community, “cross-session” or “cross-time” generalization is widely used to denote evaluation across different acquisition times of the same subject. **Adopting “cross-time” aligns our work with prevailing conventions in BCI literature**, improving clarity and comparability with prior studies.[1]
>
> Q6&Q7. Indeed, our experiments only evaluated several classic, relatively simple deep models, and we currently lack visualization of activation patterns. However, we believe that the primary contribution of our work lies in **the collection of a large EEG–visual dataset and the establishment of a multi-dimensional, multi-task benchmark**. Accordingly, in the revised version, we have added in the **Future Work** section the plan to extend evaluation to more advanced models and to include neural activation visualizations.
>
> Q8. We report the results for different granularity levels across the four tasks in **Tables 2 and 3**. Due to space constraints, we did not include representative stimulus images in the main text. However, the full list of all stimuli is available in our **code repository**, which we explicitly point to in the paper.
>
> Q9. As mentioned in our comment on Nov 13, we chose a 500 ms stimulus duration mainly to **capture semantic ERP components such as the N400**. While we acknowledge that neural responses to adjacent images might overlap, we fully separate training and testing in time. Our dataset is intentionally designed to capture rich semantic EEG responses, following a **semantically motivated experimental protocol** rather than a rapid-RSVP design (e.g. 50ms in Things-EEG dataset).
>
> Q10. The reconstruction experiments in Figure 6 of the Appendix were based on Stage 1, as cross-time and cross-subject reconstruction have not yet reached a usable level. We apologize for any unintended confusion arising from this section, even though we have explicitly elaborated on the experimental setup. In the revised version, we have **removed this part entirely from the appendix**.
>
> Thank you very much for your reviews and comments on our paper. If you have any further questions or concerns, we would be happy to continue the discussion. If you find that this response satisfactorily addresses your concerns, we kindly ask you to consider adjusting your review scores accordingly.
>
> [1] Apicella, A., Arpaia, P., D’Errico, G., Marocco, D., Mastrati, G., Moccaldi, N., & Prevete, R. (2024). Toward cross-subject and cross-session generalization in EEG-based emotion recognition: Systematic review, taxonomy, and methods. Neurocomputing, 604, 128354.

---

### Official Review · Reviewer_eNPo · 2025-10-30

**Soundness:** 3
**Presentation:** 3
**Contribution:** 2
**Rating:** 2
**Confidence:** 4

**Summary:**

This paper presents EEG-Imagenet, which is a dataset for observing image stimuli and the corresponding signals of electroencephalogram (eeg), containing 16 subjects and 4,000 images from ImageNet. In addition to the collection process of the dataset, the article also defines the evaluation methods and tasks, and demonstrates the performance of common methods in these tasks.

**Strengths:**

The motivation of the article is very good. Currently, EEG-image datasets are lacking, and the field requires datasets that include more subjects and more samples. The writing of the article is also good, easy to understand, and the several task divisions proposed (WT, PT, CT and CP) are very comprehensive and appropriate.

**Weaknesses:**

The biggest problem with the article is that the way the dataset is collected itself might be problematic. The article will continuously display the images of the same categories (50 images) to each subject, which will cause the subjects to maintain relatively stable brain activity during this period. This leads to the fact that there is actually no distinction between the images, and even brain activity simply reflects a category. This issue has been discussed in ''The Perils and Pitfalls of Block Design for EEG Classification Experiments'', IEEE TPAMI, 2020 It is generally believed within the field that this approach is incorrect.

This issue was also reflected in the subsequent experimental results, such as the WT results being very high. One possible validation is to conduct a search within images of the same category. I believe the model may not be able to retrieve effectively because continuous brain activities within the same category are likely to converge. In this case, the conclusion of the article may be affected, such as the accuracy rate of some task divisions mentioned later.

At the same time, the article does not compare this dataset with other datasets, such as when training classification models simultaneously, how the performance of the models trained on other datasets (such as THINGS) differs from that on this dataset. This is also a verification of whether the collection of the dataset is effective.

**Questions:**

Please see the Weaknesses.

---

> ### Author Response · Authors · 2025-11-25
>
> 1. **Regarding the temporal-effect concern.**
>
> We refer the reviewer to our official comment (posted on Nov 13), where we provided a detailed unified response to all reviewers. The key point is that our current experimental paradigm **does not suffer from the temporal-effect issue identified in prior work**, such as the block-design leakage discussed in Li et al. (TPAMI 2020).
>
> Importantly, even the WT task you mentioned is conducted in Stage 2, where training and testing are strictly separated across blocks. This eliminates the possibility of sampling both training and testing data from the same temporally continuous sequence, which is the root cause of the temporal-effect problem in earlier studies.
>
> The WT task shows higher performance simply because it is inherently the easiest setting (no cross-time or cross-participant generalization is required). This is fully expected and is consistent with prior EEG-category-decoding literature rather than evidence of temporal leakage.
>
> To address your concern, we have now **updated Figure 2 with more annotations and explanations**, explicitly describing how each task handles train/test partitioning and stimulus ordering to avoid temporal bias. This revision can clarify the design of our experiment and show that no temporal effect exists.
>
> 2. **Regarding comparisons with other datasets.**
>
> Metadata-level comparisons with existing datasets are provided in Table 1.
> However, **the experimental intent and paradigm design of our dataset fundamentally differ** from those of other EEG-vision datasets, which makes direct performance comparison inappropriate or even misleading.
>
> For example, THINGS-EEG, as you mentioned, contains 1700+ categories, but **only 10** images per category, and uses **a 100-ms RSVP-style** presentation.
> Such short exposures do not afford the emergence of the **N400** component, which is well-established as crucial for semantic processing (typically arising ~250–500 ms post-stimulus). Consequently, semantic object-category classification on THINGS-EEG is extremely difficult and not directly meaningful, because the paradigm is designed to study early perceptual responses, not semantic decoding.
>
> Our dataset, in contrast, is intentionally designed to capture rich semantic EEG responses and, therefore, follows a different and **semantically motivated experimental protocol**. For this reason, direct comparisons of classification performance across mismatched paradigms would not constitute a fair or informative evaluation.
>
> Thank you very much for your reviews and comments on our paper. If you have any further questions or concerns, we would be happy to continue the discussion. If you find that this response satisfactorily addresses your concerns, we kindly ask you to consider adjusting your review scores accordingly.

---

### Official Review · Reviewer_Waum · 2025-11-01

**Soundness:** 3
**Presentation:** 3
**Contribution:** 4
**Rating:** 8
**Confidence:** 3

**Summary:**

This work presents a valuable and timely resource for EEG and visual neuroscience research. The dataset is carefully designed to mitigate known issues such as block-design confounds and limited category diversity, and it includes both coarse- and fine-grained visual labels. It is thoroughly benchmarked across several classical and deep learning models, providing a solid empirical baseline for future studies. The methodology and documentation are transparent and reproducible, with clear ethical considerations and open data plans. Overall, EEG-ImageNet represents a meaningful and nice contribution that could have significant impact in advancing EEG-based decoding and facilitating cross-disciplinary work between neuroscience and machine learning.

**Strengths:**

This work presents a valuable and timely resource for EEG and visual neuroscience research. The dataset is carefully designed to mitigate known issues such as block-design confounds and limited category diversity, and it includes both coarse- and fine-grained visual labels. It is thoroughly benchmarked across several classical and deep learning models, providing a solid empirical baseline for future studies. The methodology and documentation are transparent and reproducible, with clear ethical considerations and open data plans. Overall, EEG-ImageNet represents a meaningful and nice contribution that could have significant impact in advancing EEG-based decoding and facilitating cross-disciplinary work between neuroscience and machine learning.

**Weaknesses:**

I think it would be valuable to include more extensive comparison with other dataset. It could be interest to compare the effect of having models pre-trained on other EEG datasets (e.g., SEED, DEAP, Things-EEG) and vice-versa to assess transferability and confirm that EEG-ImageNet enables broader generalization.

**Questions:**

1. Could the authors compare EEG-ImageNet with models pre-trained on other EEG datasets (e.g., Things-EEG, SEED, DEAP) to quantify transfer learning performance? This would help assess whether EEG-ImageNet generalizes beyond its own benchmark.
2. How consistent are the EEG signal qualities across sessions and participants? Were any metrics (e.g., SNR, channel dropout rates) tracked to ensure data reliability?
3. Can you give more details on the pretraining (PT) setting, it was a bit unclear to me what was done.

---

> ### Comment · Reviewer_Waum · 2025-11-27
>
> I read the other responses and the comments from other reviewers. I think that there is problems that I overlooked. I will keep my score but reduce my confidence.

---

> ### Author Response · Authors · 2025-11-29
>
> **Regarding comparisons with other datasets.**
>
> Metadata-level comparisons with existing datasets are provided in Table 1.
> However, **the experimental intent and paradigm design of our dataset fundamentally differ from those of other EEG-vision datasets**, which makes direct performance comparison inappropriate or even misleading.
>
> For example, THINGS-EEG contains 1700+ categories, but only 10 images per category, and uses a 100-ms RSVP-style presentation.
> Such short exposures do not afford the emergence of the N400 component, which is well-established as crucial for semantic processing (typically arising ~250–500 ms post-stimulus). Consequently, semantic object-category classification on THINGS-EEG is extremely difficult and not directly meaningful, because the paradigm is designed to study early perceptual responses, not semantic decoding.
>
> Our dataset, in contrast, is intentionally designed to capture rich semantic EEG responses and therefore follows a different and semantically motivated experimental protocol. For this reason, we believe that direct comparisons of classification performance across mismatched paradigms would not constitute a fair or informative evaluation.

---

### Author Response · Authors · 2025-11-13
**We are aware of the temporal effect and have ensured that no such effect exists in the experimental setup.**

Thank you very much for your reviews and comments on our paper. We noticed that several reviewers raised concerns about the issue of temporal effect (reviewer eNPo: “the way the dataset is collected itself might be problematic”; reviewer DHLu: “the structure of the experiment still presents images from the same category in short temporal clusters”; reviewer tGfm: “Thus the 2-session design does not remove the confound.”). **We would like to clarify that our research differs from Spampinato et al. and a series of research that suffered from the temporal effect.**

Spampinato et al. (2017) used a block-design experimental paradigm, and Li et al. (2020) highlighted that such block-design setups may suffer from temporal-effect bias (i.e., classification performance being driven by temporal correlations rather than semantic category information). **We did mention this issue in our manuscript (see lines 89-93, 241-243, 316-320).** At the same time, the experimental design in our paper is significantly different, and we have taken efforts to ensure that no such temporal-effect bias exists in our experiments:

1. For all tasks (WT, CT, CP, and PT), no image stimuli presented in the same block appear in both the training and testing sets.

2. We notice that some reviewers are concerned about WT. However, the training set and testing sets are collected from different blocks in stage 2. We also clarify that all tasks (including WT) are tested in stage 2 for a fair comparison, while the data in stage 1 is only for training.

3. Regarding reviewer tGfm’s concern about the work of Li et al. (2020) on subject 6 (they collected block data three times across distinct sessions and did cross-block classification): we note that their paradigm is quite different from our Stage 2 design. Actually, their paradigm is closer to our **CT** task in spirit. And they found classification performance approaching chance under that kind of cross‐block design (which is **consistent** with our findings). We used two‐way identification rather than raw accuracy in order to provide better numerical discrimination across tasks/models. We fully acknowledge the validity of their concerns, and we believe our experiment addresses these through our design choices.

4. On the suggestion of completely randomized image order, which reviewer tGfm mentions is common in RSVP paradigms (e.g., each image ~50–100 ms, very rapid presentation), (1) we argue that such **a short presentation time is not sufficient for semantic image understanding**. For example, although rapid object recognition can begin very early, recent work shows that recognizing minimal images (i.e., degraded or constrained‐view objects) still requires presentation times up to 500 ms or more to reach robust accuracy (Harari et al., 2020). More importantly, semantic processing (as indexed by ERP components such as the N400) typically occurs in the ~250–500 ms post‐stimulus window and often later when deeper semantics or scene context is involved. (2) Moreover, when images are presented in a fully shuffled and highly rapid sequence, **the neural responses to consecutive stimuli inevitably overlap**, leading to semantic interference rather than purely independent trials. In such cases, the EEG signal reflects a mixture of residual activity from previous stimuli and the onset of the current one, making it harder for models to learn consistent semantic category representations. In contrast, our paradigm—where images of the same category are presented together during training but strictly separated between training and testing—allows the brain to form a stable and **discriminative category-level representation** while still preventing temporal leakage. This design thus facilitates the decoding of true semantic category differences, rather than transient overlaps between successive stimuli.

5. A few reviewers pointed out that Figure 2’s description of the task setups was not sufficiently clear. We agree and will revise in the next version by adding more annotations and clarifications for each task to ensure that the reader fully understands how training/test splitting and stimulus ordering avoid temporal bias.

If you have any further questions or concerns about the temporal effect, we would be happy to continue the discussion. If you find that this response satisfactorily addresses your concerns about the temporal-effect issue, we kindly ask you to consider adjusting your review scores accordingly. For other questions raised, we will proceed to individual responses to each reviewer’s specific comments.

---

### Meta-Review · Area_Chair_xNKR · 2026-01-06

**Summary:**

The paper presents a benchmark dataset for EEG-based object classification in images. It includes new EEG recordings under presentation of visual stimuli, and evaluation protocols. Four classification tasks are tested using the datasets.

Strength: All reviewers basically agree with the motivation of collecting a new dataset and acknowledge the efforts for that.

Weakness: (1) Three reviewers have concerns regarding the data collection protocol: The protocol follows a block-design paradigm that can cause temporal effects. Reviewer tGfm mentions that even though training and test data are from different blocks, the block confound is not removed. Several reviewers question why randomized trials are not adopted. Reviewer eNPo mentions that since same category images are presented consecutively, the brain activity would remain stable without distinction between images.

(2) Several reviewers have concerns regarding lack of comparison to existing datasets for proving improved generalization through the proposed dataset.

(3) Several comments from multiple reviewers are about lack of details of the experiments in the paper.

(4) The work lacks of recent models and analysis of learned features or model activations (Reviewer DHLu).

**Reviewer Concerns:**

(1) The authors responded that their experiments do not have the issue of temporal effects, because the data from Stage 1 are used only for training. This explanation partly resolves the concern. The authors' rebuttal mentions that completely randomized trials may cause overlap in the neural response between consecutive stimuli. However, the block-design experiments also cause overlap neural responses between consecutive stimuli from the same category but different visual contents, as noted by Reviewer DHLu (question #9, which is not well resolved by the response). Furthermore, the concern by Reviewer eNPo (no distinction between neural responses to different images) is not resolved. Considering these, AC got fundamental questions that are still unclear in the paper: What has been actually learned by the model built with the proposed dataset, and in which practical situations the learned knowledge can be useful?

(2) The authors rebuttal mentions that experimental comparison to existing datasets is infeasible, which AC would disagree. The idea suggested by Reviewers Waum and eNPo (comparison of models trained on different datasets) could be still tested. In fact, AC feels that the paper does not showcase advantages of the proposed dataset well enough toward the ultimate goal (i.e., improving generalization capability). Furthermore, the proposed dataset is still limited in size and diversity (as noted by Reviewer DHLu), and there are existing datasets having similarly large size and no block confound (as noted by Reviewer tGfm).

(3) The authors revised the paper, which would largely clear the concerns regarding the definitions of the tasks and the data collection process.

(4) The rebuttal mentions that inclusion of recent models and analysis of learned features is left as future work. However, AC thinks that this issue needs attention in the paper to answer the central question: whether the models have learned neural correlates meaningful toward cross-participant and cross-time generalization (as the title suggests) through the proposed dataset.

**Reviewer Scores:**

AC would not expect that the reviewers would have changed their scores.

---

### Decision · Program_Chairs · 2026-01-26

Reject